# Interfacial Phenomena between Liquid Si-rich Si-Zr Alloys and Glassy Carbon

**DOI:** 10.3390/ma13051194

**Published:** 2020-03-06

**Authors:** Donatella Giuranno, Wojciech Polkowski, Grzegorz Bruzda, Artur Kudyba, Javier Narciso

**Affiliations:** 1Institute of Condensed Matter Chemistry and Technologies for Energy (ICMATE), National Research Council of Italy (CNR),Via De Marini 6, 16149 Genoa, Italy; 2Łukasiewicz Research Network - Foundry Research Institute, Zakopiańska 73 Str., 30-418 Krakow, Poland; wojciech.polkowski@iod.krakow.pl (W.P.); grzegorz.bruzda@iod.krakow.pl (G.B.); artur.kudyba@iod.krakow.pl (A.K.); 3Instituto Universitario de Materiales de Alicante (IUMA), Universidad de Alicante, Apdo. 99, 03080 Alicante, Spain; 4Departamento de Química Inorgánica, Universidad de Alicante, Apdo. 99, 03080 Alicante, Spain

**Keywords:** MMCs (Metal Matrix Composites), silicon carbide, wetting, refractory materials, Zr-silicides

## Abstract

To succeed in the design and optimization of liquid-assisted processes such as reactive infiltration for the fabrication of tailored refractory SiC/ZrSi_2_ composites, the interfacial phenomena that occur when Si-rich Si-Zr alloys are in contact with glassy carbon (GC) were investigated for the first time by the sessile drop method at T = 1450 °C. Specifically, two different Si-rich Si-Zr alloys were selected, and the obtained results in terms of wettability, spreading kinetics, reactivity, and developed interface microstructures were compared with experimental observations that were previously obtained for the liquid Si-rich, Si-Zr, near-eutectic composition (i.e., Si-10 at.%Zr) that was processed under the same operating conditions. The increase of the Si content only weakly affected the overall phenomena that were observed at the interface. From the practical point of view, this means that even Si-Zr alloys with a higher Si content, with respect to the near eutectic alloy, may be potentially used as infiltrant materials.

## 1. Introduction

Currently, SiCp (reinforced SiC particles), SiC/SiC_f_, and C_f_/SiC composites that are densified or filled by Si-based alloys and transition metal silicides are gaining great interest as thermal barriers, structural materials, and components for assembling re-entry space vehicles and fission/fusion nuclear reactors for their low density, remarkable hardness, chemical inertness, improved electrical and thermal stability, and excellent oxidation resistance, mainly at high temperatures [1,2,3,4,5,6,7,8]. For instance, refractory alloys and silicides, such as Zr-based alloys and Zr combined with Si, are already used as cladding materials for nuclear reactors due to their ultra-high neutron capture cross sections and an excellent long-term resistance under irradiation [8,9,10,11,12].

To design and produce such composites with tailored properties, the reaction bonded/formed silicon carbide process, belonging to spontaneous/pressure-less infiltration mechanisms, has well-known benefits over other conventional and costly sintering techniques, namely lower processing parameters (i.e., temperature, pressure, and time) and nearly-net shape fabrication capabilities [13,14,15,16,17]. In fact, SiC/ZrSi_2_ composites have already been successfully produced by the reactive infiltration of three different liquid Si-rich Si-Zr alloys into bi-modal SiC preforms [18]. The thermal compatibility between both phases (SiC and alloy) is excellent, as evidenced by the fact that debonding phenomena do not occur, and a low coefficient of thermal expansion (CTE) was obtained by the thermo-mechanical characterization that was performed on an as-produced SiC/ZrSi_2_ composite.

As reported in the literature, reactive infiltration has also been used to produce C_f_/ZrC–SiC composites with outstanding ablation and erosion resistance even at temperatures higher than 1600 °C. The composites were produced by infiltrating C-fiber preforms with liquid Si-8.8 at.% Zr eutectic and Si-10 at.% Zr hypo-eutectic alloys [19,20,21,22]. To limit the C-dissolution phenomenon, the C-felts were previously coated with pyrolytic C by using CVD (chemical vapor deposition). As a further confirmation, liquid metal infiltration is currently used to fabricate SiC_f_-reinforced Si-based eutectic alloy composites, which are currently under development for aerospace applications—specifically, Si-16 at.% Ti, Si-22 at.% Co, Si-38 at.% Co, Si-17 at.% Cr and Si-27 at.% Fe alloys. The mentioned Si-based alloys have been investigated as infiltrants of amorphous SiC-fibers with C as the interface layer for improving the overall oxidation resistance at high temperatures. In particular, the promising oxidation resistance can be successfully predicted by thermodynamic studies, as has been documented in [23].

In order to succeed in designing and manufacturing such composites, key know-how could come from fundamental investigations of thermodynamic and thermophysical properties of the melt phase, such as surface properties, density, and viscosity [24,25,26]. In parallel, the wetting characteristics and reactivity at the alloy/preform interfaces ought to be evaluated [27,28,29,30,31,32]. The as- collected knowledge can be profitably used as input data for optimizing the infiltration process [16,33].

In order to provide new knowledge on the interaction phenomena that occur when a liquid Si-rich Si-Zr alloy is in contact with C- or SiC-based materials, a careful experimentally-based study was done by analyzing the behavior of a liquid Si-10 at.% Zr near eutectics alloy in contact with both amorphous C (i.e., GC—glassy carbon) [31] and hot-pressed polycrystalline silicon carbide (HP-SiC) [32]. Namely, wettability, reactivity, and the overall interfacial phenomena that have been observed at the interfaces were analyzed and related to the operating conditions.

To scale up the reactive infiltration process from the lab to the industrial scale, several affecting factors should be taken into consideration. Specifically, the starting metal materials may suffer from the presence of impurities and of a more pronounced inhomogeneity in the alloy composition, usually resulting in an Si-segregation at the alloy surface that can possibly promote oxide-scale growth. Additionally, the temperature that is imposed to infiltrate the melt alloy into the preform is usually around 1450 °C, with a heating rate around 5 °C/min or even slower. Consequently, since the selected infiltrants are lower-melting Si-based alloys (i.e., near eutectic compositions), the infiltration process may start at temperatures lower than 1450 °C. Moreover, if the starting material differs from the nominal composition, not homogeneous and anomalous melting due to the presence of large amount of pure Si, is expected.

For this reason, the influence of Si content on the interfacial reactions that occur between the Si-Zr system and the GC was investigated by the contact heating sessile drop method at T = 1450 °C, and the main relevant results are reported in the present paper.

In order to check the reliability of the obtained results, the contact angle behaviors observed on the GC for the two investigated Si-rich Si-Zr alloys were compared with the experimental observations that were provided by wetting tests that were previously performed on pure Si and Si-10 at.% Zr in contact with the GC under the same operating conditions and by applying the same experimental procedure.

The microstructural characterization of the interface was performed by using light microscopy (LM) and scanning electronic microscopy (SEM) coupled with an x-ray dispersive energy analyzer (EDS). For the sake of clarity, the scope of the paper is to highlight how wetting (the key property for infiltration processes) studies may provide useful information to optimize melt infiltration in terms of operating parameters. In other words, by focusing on interfacial phenomena in terms of adhesion, reactivity, growth of reaction layers, etc., some factors negatively affecting the reactive infiltration process (i.e., pore narrowing/pore closure and/or fiber-degradation by C-dissolution) can be easily predicted and limited to a large extent.

## 2. Materials and Methods

Glassy carbon (GC) provided by Alfa-Aesar (Kandel, Germany) was used with its as-received surface conditions. A surface roughness of Ra ≈ 20 nm on a GC area of 3 × 3 mm^2^ was measured by an optical confocal-interferometric profilometer (Sensofar S-neox).

The Si-Zr alloys samples were prepared by mixing high purity (99.98%-Goodfellow®,), mechanically cleaned, and weighted Si and Zr pieces that were arc-melted under an Ar (99.9999%, Air liquid, Italy) atmosphere. In order to reduce the residual oxygen content inside the arc-melting chamber, a Zr drop was previously melted. To ensure the homogeneity of the alloy composition, each Si-Zr sample (having an initial mass of 0.05 g) was re-melted 3 times. By checking the final weight of the alloy samples, no evidence of material loss by evaporation was found. The microstructure and composition of the as-produced Si-Zr alloys at the cross-sectioned sample were analyzed by SEM/EDS (Leo 1450 VP, INCA Energy 300, Zeiss, Germany), as shown in Figure 1.

As can be seen, a two-phase of Si + SiZr_2_ that was segregated at the Si-grain boundaries was detected both in the bulk of Si-1.7 at.% Zr and Si-5.3 at.% Zr alloy samples (i.e., Si-5 wt.% Zr and Si-15 wt.% Zr, hereafter denoted as Si-5Zr and Si-15Zr). The predominance of the Si + SiZr_2_ phase depended on the Si content, as expected.

In both cases, a layer of pure Si was found to be segregated at the alloy drop surface. Contrarily, pure Si and ZrSi_2_ were detected at the surface of the Si-10 at.% Zr alloy sample (hereafter Si-27Zr). Moreover, ZrSi_2_ precipitates that were embedded into the eutectic matrix with a Zr content that varied from 8.15 to 9 at.% were revealed at the bottom of the drop.

Before the wetting experiments, the Si-Zr alloy sample and the substrate were weighted, rinsed in an ultrasonic bath, and dried with compressed air.

The Si-Zr/GC assembly was placed on a graphite sample holder, located at the central part of the heater, and leveled at the horizontal plane.

In order to remove any contaminant from the experimental environment, the device was outgassed for two hours under vacuum conditions (P_tot_ ≤ 10^−6^ mbar).

The alloy/substrate sample was heated (at the rate of 10 °C/s) by an 800 kHz high frequency generator that was coupled to a porous graphite tube [27], which provided an atmosphere with a reduced O_2_ content.

To limit evaporation phenomena, wetting experiments were performed under a static Ar atmosphere (O_2_ < 0.1 ppm), following the procedure detailed elsewhere [28].

The evolution of the contact angle values and drop geometric variables (R-base radius and H-drop height) were monitored in real-time and recorded (10 frames/sec) by a high resolution CCD-camera that was connected to a computer. Every single frame was processed by an image analysis software ad hoc-developed (ASTRAVIEW®) [34]. By analyzing the experimental method that was used and all the factors that potentially affected the reliability of the results, an accuracy of the data around ±2° was estimated [35].

After the wetting experiments, the samples were cold-embedded in an epoxy-resin (Struers©, Detroit, USA), cross-sectioned, metallographically polished, and prepared for a microstructural characterization that was performed by the LM and SEM/EDS techniques.

## 3. Results

Figure 2 shows the wetting behavior of the liquid Si in contact with the GC substrate at T = 1450 °C. In particular, by analyzing both the variation of the contact angle as a function of time and the sequence of images that were recorded during the first 25 s of the wetting test, an anomalous behavior during the Si melting was observed (Figure 2a). After 25 s from the beginning of the isothermal step, the complete melting of Si was reached, and a contact angle value of θ = 70° was measured at the Si/GC triple line. In the following 60 s, the steady state condition at the interface was achieved, and a contact angle value of θ = 38° was shown at the triple line; this angle did not change until the end of the experiment. After 15 min, the Si/GC couple was fast cooled down to room temperature (cooling rate 20 °C/s) by turning off the power to the heater, and the typical conical shape at the top of the solidifying drop was observed (Figure 2b).

Depending on the distance from the Si/GC triple line, different microstructures were developed, as shown in Figure 3. Specifically, the presence of a thin SiC layer was found close to the drop perimeter. Contrarily, moving far from the triple line, circular SiC nodules were well distinguished.

Individual SiC-crystals epitaxially developing to a size of 5–7 μm were detected at the Si/GC triple line, as shown in Figure 4. In contrast, moving to the middle of the drop, at the cross-section of the Si/GC sample, it was possible to easily identify the interface that is typical of evolving wettability by reactive mechanisms. Namely, C-dissolution pockets at the bottom and a compacted layer of SiC crystals with a size varying from 2 to 5 μm were revealed (Figure 4b).

In Figure 5, the evolution over time of the contact angle for the Si-5Zr/GC and Si-15Zr/GC systems are shown, and they are compared with the wetting behavior that was displayed by the liquid Si-27Zr alloy in contact with the same substrate under the same operating conditions [31].

As it can be observed, it seems that the achievement of the equilibrium conditions were not affected by the different Si contents, and the wetting kinetics almost overlapped (Figure 5b). However, a slight influence of the Si content in the final contact angle value deserves to be pointed out (Figure 5a). Specifically, contact angle values of θ = 44, 41, and 38° were measured after 15 min at the Si-27Zr/GC, Si-15Zr/GC and Si-5Zr/GC triple lines, respectively. It is worth pointing out the irregular wetting behavior, in respect to the other investigated two systems, that was exhibited by the Si-27Zr/GC system within the first 40 s, as it is shown in Figure 5b.

In Figure 6, SEM/EDS by back-scattered electron (BSE) analyses that were performed both at the top drops and at the triple lines of the Si-5Zr/GC (Figure 6a,b) and Si-15Zr/GC (Figure 6d,e) samples that were processed at T = 1450 °C for 15 minutes are shown. The presence of SiC-crystals with a size ranging from 2 to 10 μm was found to delimit the drops perimeter. Similarly to the Si/GC couple that was processed at the same testing temperature, a thin layer of SiC surrounding the drop perimeter at the triple line was found. In particular, moving far from the triple line, unreacted regions of GC with over-layered circular and narrowing areas of SiC are well distinguished. Moreover, owing to the applied fast cooling of the samples at the end of the wetting experiment, a crack at the interface was also observed.

Though fragmented, a compact layer of SiC with a thickness of 5–7 μm that was grown both at the triple lines (Figure 6b,c) was detected. Moving toward the middle of the drop, different microstructures were observed, as shown in Figure 6c,f. In particular, individual SiC-crystals and C-dissolution pockets were revealed at the Si-15Zr/GC interface. Contrarily, an alternating microstructure of thick compact SiC-layers and C-dissolution pockets in contact with the Si-Zr alloy were found at Si-5Zr/GC system.

## 4. Discussion

As pointed out in the previous section, anomalous wetting kinetics, due to a delay in achieving the complete melting of the sample at the liquid Si/GC interface, was observed (Figure 2). Such behavior is typically noticed during wetting tests that are performed by contact heating sessile drop experiments [35], and it is caused by the presence of an SiO_2_-native oxide layer at the metal surface. However, the native oxide should be decomposed by the following Reaction (1) [36]:SiO_2(s)_ + Si_(l)_ → 2SiO_(g)_ ↑(1)

When the SiO_2_-layer completely disappears at the interface, it enables the liquid Si to interact with the GC substrate, according to the following exothermic Reaction (2):Si_(l)_ + C_(s)_ → SiC_(s)_(2)

Indeed, a compact layer of SiC was found at the interface (Figure 4). Additionally, the packaging of SiC-crystals at the interface was already achieved in 15 minutes, and this prevented a further interaction phenomena between unreacted C and liquid Si. On the basis of such evidences, the equilibrium at the interface can be assumed.

After 85 s from the beginning of the isothermal time, the measured contact angle value of θ = 38° was in a very good agreement with the literature data that have been reported for the liquid Si in contact with the amorphous C, and even with SiC [37]. In this regard, the wetting of C by liquid Si and Si-based alloys was driven by a combination of reactive and no-reactive mechanisms, as described in [36]. In addition, the equilibrium contact angle that was exhibited at the triple line actually resulted from the interaction phenomena that occurred between the metal phase and the already produced interfacial SiC layer (Reaction (2)). Moreover, as widely explained in [38], the wettability of liquid Si and Si-based alloys in contact with C-materials evolved through different spreading stages. Within the first stage (lasting a few seconds), the C-dissolution and reaction between C and liquid Si, resulting in the appearance of an SiC-thin layer, were the main interaction phenomena that took place at the interface. Subsequently, a decrease in the reactivity was observed due to the aggregation of individual SiC-crystals that were grown at the interface and actually impeded the direct contact between the reacting phases (i.e., liquid Si and C).

The further advancing of the triple line was favored by the growth of a thin layer of SiC beyond the drop perimeter. It resulted from the evaporation/condensation phenomena of the liquid Si onto the unreacted C substrate, as shown in Figure 3.

It is worth noting that the final contact angle values that are reported in the present work are in good agreement with the value measured at the Si-27Zr/SiC and with the contact angle shown by the dispended drop method [30]. Such outcome let us to conclude that the wetting kinetics and related interfacial phenomena occurring at the Si/GC interface were affected by the presence of SiO_2_ that was segregated at the interface, only during the early stage of the experiment.

As already introduced, a previous study focused on the wetting characteristics of liquid Si-27Zr in contact with GC [31] and SiC [30] as a function of operating conditions (e.g., temperature and time), and the method that was applied there was used here. Namely, the interfacial phenomena that were observed in terms of contact angle behavior, reactivity, spreading kinetics, and microstructural evolution, were analyzed.

In Table 1, the final contact angle values (θ_f_) and spreading rates (U_spread_) that were observed at the triple lines as functions of temperature and time are reported and compared with the results provided in the present work.

The interfacial phenomena that occur between liquid Si-27Zr alloy and its GC substrate were widely described in [31]. In summary, the reactive wetting was defined as the main interaction mechanism that takes place when a liquid Si-Zr alloy gets in contact with amorphous C. This is confirmed by analyzing the spreading rates reported in the Table 1 and Figure 7. Specifically, it was obtained that U_spread_ (1354 °C) < U_spread_ (1400 °C) < U_spread_ (1450 °C), as expected. Taking into account the final contact angle value that was exhibited after 15 minutes between the liquid Si-Zr alloy and the GC, the fact that θ_f_ (1354 °C) > θ_f_ (1400 °C) > θ_f_ (1450 °C) is also not surprising, as shown in Figure 7b. The predominance of reactivity as the mechanism that controlled the wettability between the liquid Si-27Zr alloy and the GC was also confirmed by the value of the activation energy E_a_ = 222 kJ/mol, that was calculated by using the Arrhenius plot of the spreading velocity (Figure 8a), which typically describes reactive mechanisms [36].

The results concerning the interaction phenomena that were observed at the Si-27Zr/GC interface as a function of the operating overviewed conditions above, mainly in terms of spreading rates and measured final contact angle values, can be easily explained by taking into account the fact that all the kinetics related to reactive, diffusion and dissolution phenomena, as well as the evaporation tendency of the liquid metal phase, are enhanced when temperature is increased.

The influence of the testing temperature that was selected on the final contact angle value was even found at the Si-27Zr/SiC triple line, as shown in Figure 7c. Additionally, as compared to the behavior that was observed for the GC under the same operating conditions, the resulting U_spread_ (GC) < U_spread_ (SiC) is further evidence that the GC is “less” wettable than SiC. Moreover, the θ_f_ (SiC) < θ_f_ (GC) was caused by the rougher surface that was produced by the reaction at the Si-27Zr/GC interface. In particular, the “de-wetting phenomenon” was more evident for times longer than 15 min (45 min), and the increase of the size of the interfacial SiC-crystals from 2–5 to 2–10 μm determined the increase of the final contact angle value from 43 to 45°, as described in [30]. 

By taking into account the interfacial phenomena and the appearance of SiC as the reaction interlayer between Si-Zr alloy and GC before the achievement of equilibrium conditions, the contact angle values that were measured at the Si-27Zr/GC interface are comparable and in a very good agreement with the literature data. In fact, considering the little difference in Zr content, the final/equilibrium contact angle values that were measured at the Si-27Zr/GC triple lines as a function of temperature were in agreement with the values that were measured for the Si-22Zr/GC system (i.e., Si-8 at.% Zr/GC) under an Ar atmosphere [32]. Similarly, the reported contact angle values were weakly influenced by the increasing of the temperature from 1404 °C (as it came from the detected Si-Zr/GC melting point) to 1450 °C. Furthermore, during the heating of the sample, the contact angle value increased from 44° to 45°. This can be justified by the growth of SiC crystals at the interface, which determined the increase of the roughness at the GC surface.

At T = 1452 °C, the contact angle value suddenly decreased from 45° to 29°, and then it remained constant throughout the experiment at 1500 °C. In agreement with the authors’ interpretation, the interaction phenomena that occurred at the interface were most probably affected by the surface oxidation. In fact, during the wetting experiments, the Si-Zr alloy/substrate couples were surrounded by an Al_2_O_3_ tube which may have started to be more permeable and to release oxygen at high temperatures [39]. 

On the other hand, as mentioned above, under an atmosphere of reduced O_2_ content, further oxidation and O-transport phenomena at the alloy surface should be suppressed by the presence of counter flows of volatile SiO [40,41]. In addition, the appearance of stable oxides at the surface is also limited by the high evaporation tendency of liquid Si [42]. Condensed Si and Si-oxide “fogs,” acting as barriers by rejecting the O_2_-flow coming from the surrounding gas environment to the alloy surface, may be present.

In the present work, the samples were heated by induction and surrounded by a graphite tube, as described in the Section 2. Though the experiments were performed under an Ar static atmosphere, the residual oxygen content (<0.1 ppm) was further decreased by the following reactions set:C _(s)_ + O_2__(g)_ → CO_2__(g)_(3)
2C _(s)_ + O_2__(g)_ → 2CO _(g)_(4)
2CO _(g)_ + O_2__(g)_ → 2CO_2__(g)_(5)
CO_2__(g)_ + C_(s)_ → 2CO _(g)_(6)

In particular, under thermodynamic considerations, at a testing temperature of 1450 °C, Reaction (4) is amongst the most favored ones. In addition, if the mentioned reactions are concomitant with the release of SiO, typically occurring during SiO_2_ decomposition, the following reactions may take place: SiO _(g)_ + CO _(g)_ → SiC_(s)_ + O_2__(g)_(7)
SiO (g) + 3CO (g) → SiC(s) + 2CO2 (g) (8)
SiO _(g)_ + 2C_(s)_ → SiC_(s)_ + CO_2__(g)_(9)

The reactions listed above belong to the set of equilibria that are responsible of the growth of SiC micro- and nanowires on C and SiC (seeds) via a vapor–solid process, as reported in [43,44,45,46]. This type of growth is widely used to produce high quality SiC whiskers. The growth of SiC-crystals along the [11] direction results from the spiral-overlayering of hexagonal SiC planes. The final SiC-wire morphology strongly depends on the process parameters, specifically temperature, surrounding atmosphere, and starting materials, which, in other words, control the SiO-supersaturation level into the vapor phase. The temperature that is imposed to promote the growth from nano- to micro-SiC wires typically ranges from 1400 to 1600 °С. In addition, depending on the operating conditions, the process time takes at least 30 min in order to detect the appearance of the first measurable SiC nanowires.

In the present case, the presence of SiC micro-crystals that were grown up at the surface of epitaxial SiC crystals, generated in turn from Reaction (2), were detected at the triple line of the Si-27Zr/GC sample that was processed for 45 min, as shown in Figure 8a,b.

The appearance of SiC-wires at the alloy drop surface confirmed that, within the performed wetting experiments, the sample was under the deoxidation condition, since the increase of SiO partial pressure into the experimental environment (by Reaction (2)), combined with the presence of CO, was high enough to promote the vapor-solid process for growing the detected SiC nano/micro-crystals.

The collected results on wetting characteristics and spreading kinetics that are reported in Table 1 are consistent with the effect of Si content.

As pointed out in the previous section, the wetting behavior that was exhibited by the Si-27Zr/GC system within the first 40 s differed from the other two systems that were investigated, as shown in Figure 5b. The irregular wetting kinetics may have been caused by the presence of ZrSi_2_ precipitates (Tm ZrSi_2_ = 1620 °C [47]) at the bottom of the sample (Figure 1f) that were embedded into the eutectic matrix and even at the surface of alloy drop.

The weak influence of the alloy composition on the spreading rate may be consistent if the very limited range of Si content (i.e., 0–10 at.%) is considered. This can be also evinced by the comparable value of spreading rates that were observed at T = 1450 °C. However, on the measured wetting characteristics, a “more pronounced” effect of the Si content can be found in terms of the measured final contact angle value and, consequently, the trend of wettability seemed to be improved by increasing the Si content into the alloy.

According to the thermodynamic calculation of the C-Zr-Si ternary phase diagram that was reported in [39], Si-rich Si-Zr alloys with a Zr-molar fraction up to X = 0.17 should be in equilibrium with Si and SiC phases. In fact, except the presence of SiC at the interface, no other compounds were detected as reaction products, as shown in Figure 9. In addition, by comparing the SEM/EDS analyses that were performed on the cross-sectioned samples, similar developed microstructures were found at the Si-5Zr/GC and Si-27Zr/GC interfaces, both in the middle of the sample and at the triple lines. In particular, a more compacted SiC layer at the triple line was found, as expected [36]. Contrarily, in the middle of the drop, the equilibrium conditions were not yet achieved. Indeed, the presence of well-distinguished C-dissolution pockets were still noted. The different behavior that was exhibited at the interface by the liquid Si-Zr alloys that were processed at T = 1450 °C with respect to the pure Si, was most probably caused by the presence of Zr as an alloying element.

Finally, the results that were obtained in terms of a slight composition-dependency regarding the wetting characteristic, and on the interface-developed microstructures that were obtained under the same experimental conditions were in good agreement with similar investigations that were performed on other Si-based systems by the authors [27,29].

In the view of providing new knowledge for optimizing the infiltration process that is used to fabricate SiC/ZrSi_2_ composites, liquid Si-Zr alloys that are enriched with Si, with respect to the Si-27Zr alloy (usually selected as infiltrant Si-rich Si-Zr alloy), may not make the pore closure phenomenon more favorable. On the other hand, the use of Si-Zr alloys with a higher Si content should be limited to avoid the decrease of the overall thermo-mechanical response of the produced composite.

## 5. Conclusions

A comprehensive study of the interaction phenomena that occur when a liquid Si-rich Si-Zr alloy is in contact with amorphous C was carried out, and the more relevant findings are provided. In particular, a careful analysis of the contact angle behaviors over time, spreading kinetics, reactivity and interfacially-developed microstructures as a function of the alloy composition at T = 1450 °C was carried out.

It was documented that the wettability of GC by Si-rich Si-Zr alloys is controlled by a reactive mechanism.

Despite the pronounced reactivity, the wetting characteristics are only slightly composition-dependent, and the overall phenomena are controlled by the growing and thickening of SiC crystals at the interface.

The wetting characteristics and spreading kinetics that were observed at the Si-rich, Si-Zr alloys/GC triple lines at T = 1450 °C are in very good agreement with the results that were obtained from previous investigations that were performed on Si-27Zr/GC and Si-27/SiC systems.

Aiming to provide knowledge for optimizing the infiltration process that is used to fabricate SiC/ZrSi_2_ composites, liquid Si-Zr alloys that are enriched with Si, with respect to the Si-27Zr alloy that is usually selected as the infiltrating metal material, may not enhance the pore closure phenomenon. On the other hand, the use of Si-Zr alloys with an Si content larger than that of the eutectic composition should be avoided for preserving the overall thermo-mechanical response of the produced composite.

## Figures and Tables

**Figure 1 materials-13-01194-f001:**
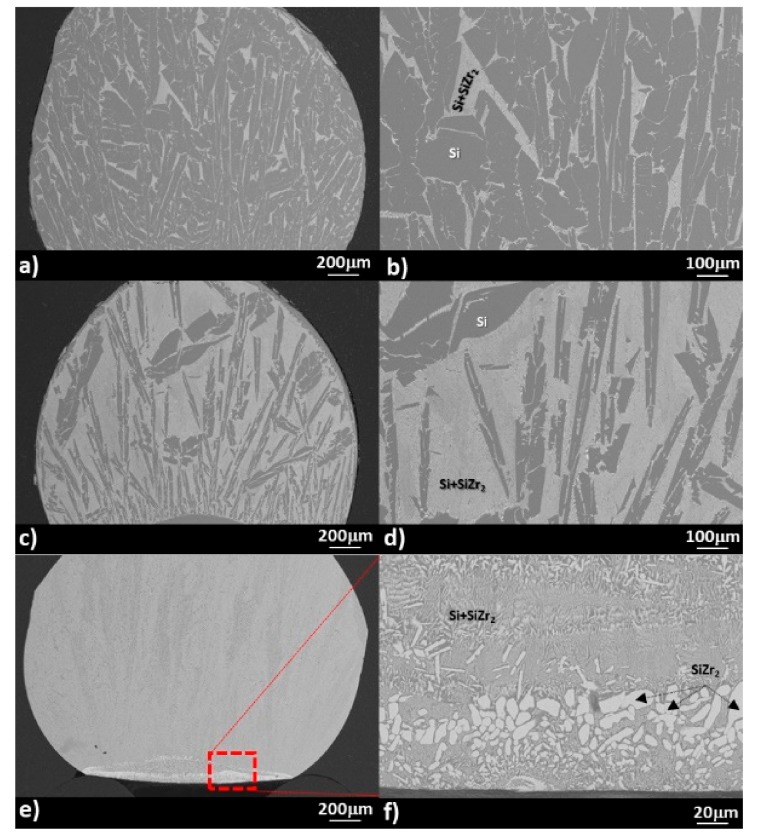
SEM/BSE (back-scattered electron) images and proposed phase identification by EDS analyses that were performed at two magnifications of the cross-sectioned as-produced alloys samples: (**a**) and (**b**) Si-5Zr; (**c**) and (**d**) Si-15Zr; and (**e**) and (**f**) Si-27Zr.

**Figure 2 materials-13-01194-f002:**
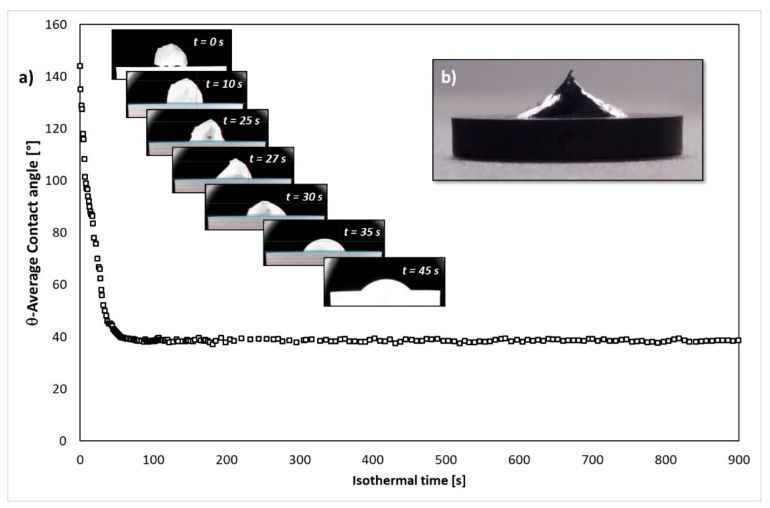
Wetting kinetics for the liquid Si/GC (glassy carbon) system observed at T = 1450 °C under a static Ar atmosphere for 15 min: (**a**) contact angle values and time sequence images of the liquid Si-drop spreading on GC; (**b**) the Si/GC sample after the wetting experiment.

**Figure 3 materials-13-01194-f003:**
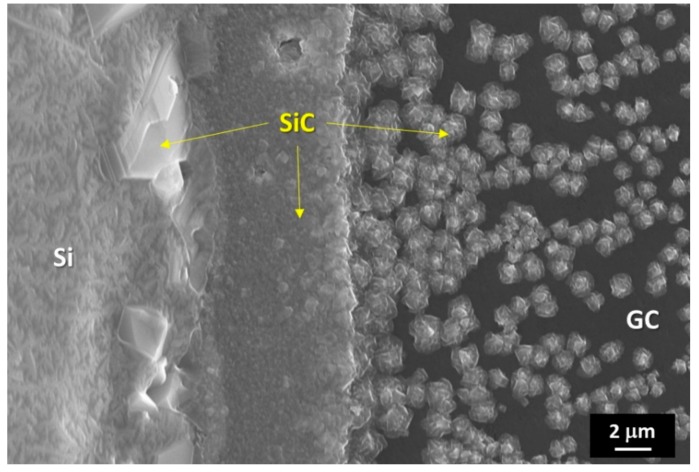
SEM image and proposed phase identification by EDS analysis that was performed on the Si/GC couple at the triple line-top drop and after the wetting test performed at T = 1450 °C under a static Ar atmosphere for 15 min.

**Figure 4 materials-13-01194-f004:**
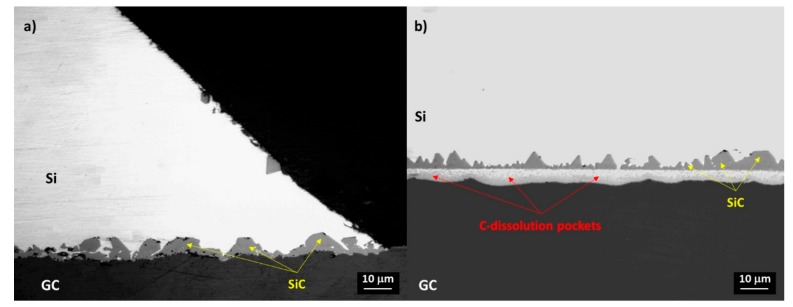
Light microscopy (LM) images at the cross-sectioned Si/GC sample after the wetting test performed at T = 1450 °C under a static Ar atmosphere for 15 min: (**a**) triple line; (**b**) middle of the sample.

**Figure 5 materials-13-01194-f005:**
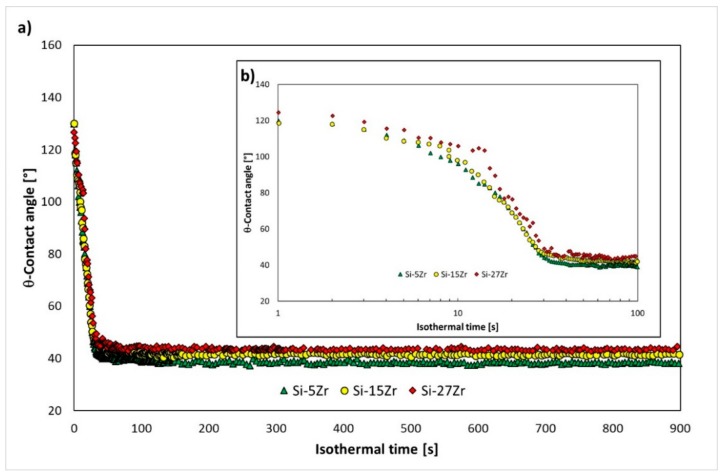
Contact angle behaviors observed at T = 1450 °C under an Ar atmosphere for (**a**) 900 s and (**b**) 100 s as a function of Si content: (◆) Si-27Zr/GC, (●) Si-15Zr/GC, and (▲) Si-5Zr/GC.

**Figure 6 materials-13-01194-f006:**
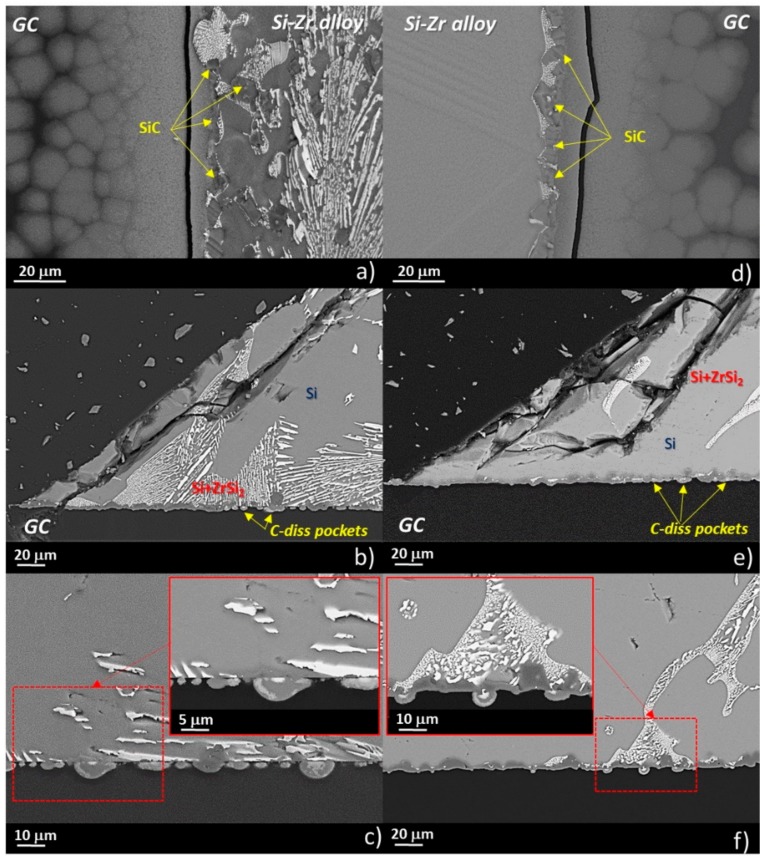
SEM-BSE and proposed phase identification by EDS analyses that were performed at different magnifications at the triple lines and at the interfaces of (**a**), (**b**), and (**c**) the Si-15Zr/GC sample; and (**d**), (**e**), and (**f**) the Si-5Zr sample after the wetting tests that were performed at T = 1450 °C for 15 min.

**Figure 7 materials-13-01194-f007:**
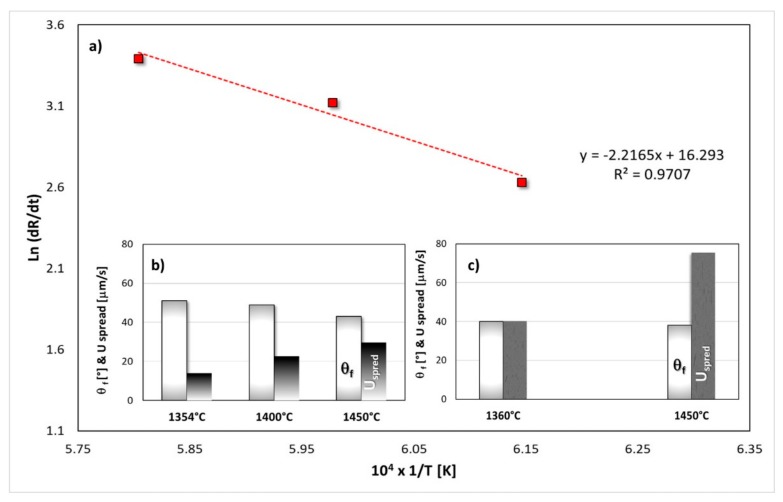
Spreading kinetics in the liquid Si-27Zr/GC system: (**a**) Arrhenius plot of the spreading rates that were observed at different testing temperatures; spreading rates and equilibrium contact angle values as a function of the temperature in the (**b**) Si-27Zr/GC and (**c**) Si-27Zr/SiC systems.

**Figure 8 materials-13-01194-f008:**
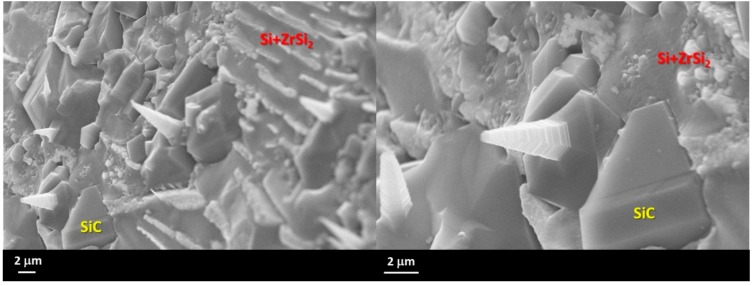
SEM images at two different magnifications and proposed phase identification by EDS analyses that were performed near the triple line of the Si-27Zr/GC sample after the wetting tests that were performed at T = 1450 °C for 45 minutes. (**a**) left; (**b**) right

**Figure 9 materials-13-01194-f009:**
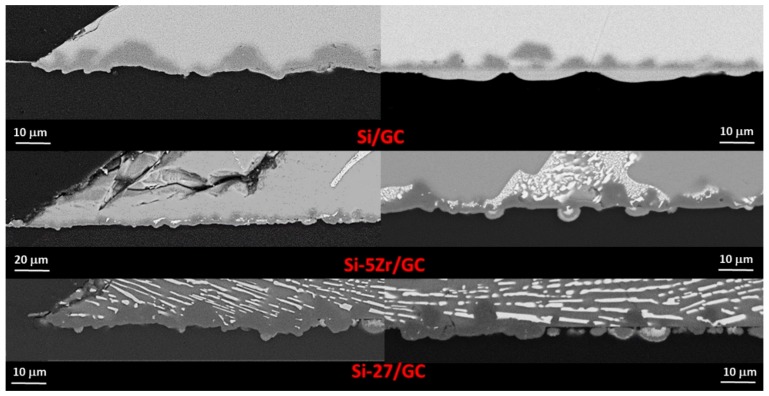
SEM-BSE images at different magnifications and proposed phase identification by EDS analyses that were performed at (left side) the triple lines and at (right side) the interfaces close to the middle of the Si/GC, Si-5Zr/GC and Si-27Zr/GC samples after the wetting tests that were performed at T = 1450 °C for 15 min.

**Table 1 materials-13-01194-t001:** Equilibrium contact angle values (θ_f_) and spreading rates (U_spread_) that were measured for the liquid Si-rich Si-Zr in contact with GC and SiC as a function of temperature and time by the contact heating sessile drop method: (*) [30] and (**) [31].

System	T (°C)	t (min)	θ_f_ ± 2 (°)	U_spread_ (μm/s)
Si-27Zr/GC**	1354	15	51-52	13.9
	1400	15	49	22.7
	1450	15	43	29.7
	1450	45	45	28.6
Si-27Zr/SiC*	1360	15	40	40
	1450	15	38	75
Si-15Zr/GC	1450	15	41	36.4
Si-5Zr/GC	1450	15	38	37.7
Si/GC	1450	15	38	26

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
