# Peer review of "Interfacial Phenomena between Liquid Si-rich Si-Zr Alloys and Glassy Carbon"

_materials, 2020, doi:10.3390/ma13051194_

Round 1

Reviewer 1 Report

In the abstract the objective should be identified. What is exactly the novelty of the paper? Why the choose of 1450 ºC? Based on what? Line 102 – 0,05 g should be replaced by 0.05 g. Line 102 – Page 102 – Why? Based on what? How many test were done? Standard deviation? Nothing is said about the equipment used. SEM; X-Ray … Line 112 – And this is good or not? What is the influence that these precipitates can have? Line 147 – Based on some standard? Line 164 – What epoxy- resin? Properties? Tg? Line 322 – Why? Based on what can the authors conclude this? Line 367 – However, no explanation has been given. Figure 7 – Only 3 points!? Line 398 - Can the authors confirm this? Or discuss with the open literature? Line 432 – References? Figure 8 – EDS results? Line 460 - “… reasonable…” Quantify? What does reasonable authors mean? Line 477 – Replace “he” by “the” !? Line 485 – “Thee”? Lines 513-515 – How can the authors conclude these? Line 513 - “… slightly…” Quantify? What does slightly authors mean?

Author Response

Dear Reviewer,
I have made a joint response to all the reviewers that you can review in the attached document.
Thank you very much for your time and your valuable comments.

Reviewer 2 Report

Giuranno et al reported the interaction phenomena when liquid Si or Si-Zr alloy was in contact with Glassy Carbon. The research is original and of practical significance and is recommended to publish with the following revision.

The language throughout the manuscript needs improvements. The term GC should be defined at its first shown place. Line 174, what is the time for contact angle value at 70° Line 177. fast cooled down to the room temperature: quenching in water? please specify. Line 178, what is the contact angle after solidifying? Figure 3 and 5. I did not see an EDS result. Only a SEM image. How did the author confirm it is SiC? the elemental composition should be provided. Line 243. Figure 6b should be figure 5b. Line 342, SiO2 did the authors see O element from EDS?

Author Response

(The authors gave the same response as above.)

Reviewer 3 Report

The manuscript reports on the study interfacial phenomena between liquid Si-rich Si-Zr alloys and glassy carbon.

The topic is appropriate for the journal.

The title is adequate and correlate with the content of the article.

The abstract reports a consistent summary of the article findings.

The work has a clear structure and the results support and test the hypothesis approprialtely.

All sections are properly written and required for a complete understanding.

Author Response

(The authors gave the same response as above.)

Round 2

Reviewer 1 Report

The authors answered of almost all the questions but did not include this information in the article.

Author Response

Dear Reviewer,

We have sent the paper to a professional service to modify it, and improve English. I thank you again for your valuable comments.
We attach the new version

Reviewer 2 Report

Most of my concerns have been addressed. I recommend it to be published.

Author Response

(The authors gave the same response as above.)
